# Allocation Mechanisms of Non-Structural Carbohydrates of *Robinia pseudoacacia* L. Seedlings in Response to Drought and Waterlogging

**Bin Yang** [1] , **Changhui Peng** [1,2,*], **Sandy P. Harrison** [1,3], **Hua Wei** [1], **Han Wang** [1], **Qiuan Zhu** [1] **and Meng Wang** [4]

[1]  Center for Ecological Forecasting and Global Change, College of Forestry, Northwest A&F University, Yangling 712100, Shaanxi, China; yangbin0729@gmail.com (B.Y.); s.p.harrison@reading.ac.uk (S.P.H.); weihua407517990@126.com (H.W.); wanghan_sci@yahoo.com (H.W.); qiuan.zhu@gmail.com (Q.Z.)

[2]  Department of Biology Sciences, Institute of Environment Sciences, University of Quebec at Montreal, C.P. 8888, Succ. Centre-Ville, Montreal, QC H3C 3P8, Canada

[3]  Department of Geography and Environmental Science, University of Reading, Whiteknights, Reading RG6 6AH, UK

[4]  State Environmental Protection Key Laboratory of Wetland Ecology and Vegetation Restoration, Institute for Peat and Mire Research, Northeast Normal University, Changchun 130024, China; mwang@nwsuaf.edu.cn

*  Correspondence: peng.changhui@uqam.ca; Tel.: +86-29-8708-0608

**Abstract:** Climate change is likely to lead to an increased frequency of droughts and floods, both of which are implicated in large-scale carbon allocation and tree mortality worldwide. Non-structural carbohydrates (NSCs) play an important role in tree survival under stress, but how NSC allocation changes in response to drought or waterlogging is still unclear. We measured soluble sugars (SS) and starch in leaves, twigs, stems and roots of *Robinia pseudoacacia* L. seedlings that had been subjected to a gradient in soil water availability from extreme drought to waterlogged conditions for a period of 30 days. Starch concentrations decreased and SS concentrations increased in tissues of *R. pseudoacacia* seedlings, such that the ratio of SS to starch showed a progressive increase under both drought and waterlogging stress. The strength of the response is asymmetric, with the largest increase occurring under extreme drought. While the increase in SS concentration in response to extreme drought is the largest in roots, the increase in the ratio of SS to starch is the largest in leaves. Individual components of SS showed different responses to drought and waterlogging across tissues: glucose concentrations increased significantly with drought in all tissues but showed little response to waterlogging in twigs and stems; sucrose and fructose concentrations showed marked increases in leaves and roots in response to drought but a greater response to drought and waterlogging in stems and twigs. These changes are broadly compatible with the roles of individual SS under conditions of water stress. While it is important to consider the role of NSC in buffering trees against mortality under stress, modelling this behaviour is unlikely to be successful unless it accounts for different responses within organs and the type of stress involved.

**Keywords:** carbon allocation; drought; non-structural carbohydrate; pools; tree mortality; waterlogging

## 1. Introduction

Extreme weather events, such as summer drought or heavy precipitation and flooding, are expected to occur more frequently in the future as a result of anthropogenic climate change [1,2]. Widespread drought-induced tree mortality has been widely documented in recent years [3–6], and more such events are anticipated by climate models in the future [7,8]. It is important to

understand climate-induced tree mortality events, both from a biodiversity and a carbon cycle perspective [8,9], but knowledge about the mechanisms involved in buffering trees against both drought and waterlogging is currently limited [10,11].

Non-structural carbohydrates (NSCs), consisting of soluble sugars (e.g., fructose, glucose and sucrose) and immobile starch and lipids [11], are produced in photosynthesis and provide the energetic basis for metabolism and growth. Isotopic studies indicate that most NSCs are used for respiration [12], providing a rapid link between assimilation, allocation, metabolism, and defence mechanisms [13,14]. However, they also act as a crucial reserve that can be used in times of stress [15,16]. There is evidence that NSCs can be stored for many years: Richardson et al. [17] found that the NSCs in the trunks of temperate forest trees were, on average, 7–14 years old. Some of this stored material may be quarantined and unavailable for future utilization [18], but other studies show that stored NSCs can be re-mobilized and used during times of stress [19–22]. Furthermore, it has been shown that NSCs allocated to leaves and fruits have been derived from recently assimilated NSCs [23,24]. Existing studies suggest that there may be 'fast' and 'slow' NSC recycling pools [12,17,25,26], and there is also evidence for the rapid mixing between these pools [27,28]. Nevertheless, there is still much to be learned about the mechanisms of NSC allocation to different structural components of the plant, the different forms of NSC, and changes in allocation and recycling under stress conditions [29].

NSCs have been used as an indicator to study the causes and mechanisms of drought-induced carbon allocation, tree growth and mortality [30]. There is evidence that NSCs allocated to roots [20,21] and stems [22] could buffer against droughts and waterlogging stresses in future climate scenarios. However, the physiological NSC allocation mechanisms specifically associated with responses to different levels of drought are still poorly known [7,31–33]. For example, a recent study by Adams et al. [34] suggested that hydraulic failure appeared more important than NSC reserves [34].

Here, we address the changes in NSC allocation and storage in response to drought and waterlogging stress, focusing on an important pioneer and drought-resistant species in arid and semi-arid regions of China: *Robinia pseudoacacia* L. (black locust). We measured and quantified changes in the NSC pools of leaves, twigs, stems and roots in response to multiple levels of water stress to address three questions: (1) How much do the amounts of soluble sugars and starch change under stress? (2) Are these responses similar in different components of the tree under droughts and waterlogging conditions? (3) Do stepwise reductions in soil moisture from FC result in synonymous stepwise decreases in biomass and starch and increases in soluble sugar?

## 2. Materials and Methods

### 2.1. Experimental Design

NSC measurements were conducted on two-year-old *R. pseudoacacia* L. seedlings grown in a greenhouse in two consecutive years, 2015 and 2016. A total of 132 seedlings were used in 2015, and a total of 88 seedlings were used in 2016. We used a randomized block design with 6 and 4 replicates in 2015 and 2016, respectively (Figures S3 and S4). The seedlings were grown in 30 cm diameter and 40 cm deep pots. Each pot contained a 1:5 ratio mixture of nutrient-poor sand and local field soil (Cumulic Anthrosol). The local soil was sieved (4-mm sieve) to remove roots, coarse organic matter and coarse sand prior to mixing. We fertilized all pots with one-quarter strength Hoagland solution [35] after transplantation to the pots. The seedlings were grown for 4.5 months prior to the beginning of the water stress treatments to minimize biases due to the initial weight and/or height of the seedlings.

We subjected the seedlings to 7 different soil moisture treatments (Table S3), encompassing a range from extreme drought (no watering, Ext) to permanently waterlogged (WL), with interim stages at water-holding capacities of 10% (W10), 30% (W30), 50% (W50), 70% (W70) and field capacity (FC). Field capacity was determined by cutting ring methods mixed with weighing pots until a constant weight was reached [36], and add ~40% more water as the waterlogged treatment. Waterlogged conditions were maintained by submerging the pots in buckets. Seedlings were randomly assigned

to different treatments. The water treatments were started on August 1 of each year and maintained for 30 days under ambient light conditions prior to the measurement of NSCs. We mainly reported and discussed the resistance of NSCs to drought and waterlogging at 10 days after the treatment in both 2015 and 2016 because the seedlings showed the strongest response to drought (especially under Ext and W10) and waterlogging (WL) at the 10-day sampling period among the three sampling periods (10-day, 20-day and 30-day), when the plant morphology showed strong wilting and foliage damage. Reflectometer probes (ML3 ThetaProbe Soil Moisture Sensor; Delta-T Devices Ltd, Cambridge, United Kingdom) were used to monitor moisture content, and the pots were watered whenever the water content varied by >1% of the target level. The seedlings in the Ext experiment underwent wilting, leaf yellowing and defoliation during the 30-day treatment. We used the average soil water content (SWC), which was measured gravimetrically at the end of the experiment, to infer the overall water status of this treatment group (Table S3).

## 2.2. Non-Structural Carbohydrate Measurements

NSC measurements were made at the end of every 10 days sampled on one randomly selected seedling from each water treatment from each block with a total of 6 seedlings in 2015 and 4 seedlings in 2016 (Figure S3). We sampled leaves, twigs, stems and roots separately on each seedling. The total concentration of NSC is defined as the sum of SS (fructose, glucose and sucrose) and starch concentrations.

SS concentrations were measured on three 1 g dry biomass samples (i.e., triplicates). SS was extracted using a warm 80% ethanol solution for 30 min and centrifugation at $5000\times g$ for 10 min; this procedure was repeated 3 times [37–40]. The supernatant was filtered through a 0.22 μm PES membrane (Sartorius Stedim Biotech, Göttingen, Germany). Afterwards, the filtrate was diluted 10 times before direct injection into the High-Performance Liquid Chromatography apparatus (HPLC-1260, Agilent, Santa Clara, CA, USA). Determination and quantification of each specific reducing sugar, fructose, glucose and sucrose, were performed using a COL–AMINO 150 × 4.6 mm column (Agilent, Santa Clara, CA, USA). The analysis was performed at 35 °C with a flow rate of 1 mL min$^{-1}$ using isocratic elution with 80% acetonitrile and 20% deionized water mixture as the mobile phase.

The starch concentration was determined using the perchloric acid method [38]. Briefly, starch was extracted from the ethanol-insoluble residues after ethanol was first removed by distillation extraction for SS. The starch in the residue was then released by boiling in 10 mL of deionized water for 15 min. After cooling to room temperature, 10 mL of 9.2 M HClO$_4$ was added and shaken for 15 min. The mixture was centrifuged at $5000\times g$ for 10 min, and the supernatant was collected. A further extraction was carried out with 10 mL of 4.6 M HClO$_4$. The supernatant was also retained, combined, and stored at $-20$ °C for starch determination based on the absorbance at 620 nm using a spectrophotometer (SMB80-3003-76, Sartorius stedim biotech, Göttingen, Germany) [41]. The total NSC pool (TNSC) was calculated as dry weight biomass (g) × NSC concentrations (g g$^{-1}$).

## 2.3. Soil Moisture and Seedling Biomass

The gravimetric soil water content was determined on a 20 g sample of sieved (2 mm mesh) and homogenized soil from each pot by oven-drying at 105 °C for 24 h. The biomass of leaves, twigs, stems and roots was estimated separately for all seedlings for each batch of samples in two years. The material was oven-dried for at least 48 h at 70 °C in paper bags (12 × 15 cm) to constant weight and then weighed.

## 2.4. Statistical Analysis

We used one-way analysis of variance (ANOVA) to examine the relationship between SWC and dependent variables among water treatments at the time of harvest. We tested for data normality and homogeneity of variance using the Shapiro-Wilk [42] and Levene tests [43], respectively. To test the

difference in SS and starch among water treatments, we used a two-way ANOVA with water treatment as a fixed factor and block and "year" as random variables. We included total seedling biomass as a covariate to account for scaling relationships between plant size and response variables [44]. We compared the mean values for each water status and treatment combination with the FC experiment. The family-wise type I error rate for contrasts was controlled at a = 0.05 using a Bonferroni correction. If there was no interaction, a post hoc multiple comparison of means was carried out with Tukey's honestly significant difference (HSD) test [45]. The results were considered significant when $p \leq 0.05$. All statistical analyses were performed using SPSS v.21.0 (IBM Inc., Armonk, NY, USA).

## 3. Results

The total biomass at the beginning of the experiments was 14.71 $\pm$ 0.50 g in 2015 and 17.80 $\pm$ 0.42 g in 2016 (Figure S4a). There was a small increase in biomass over the 10-day period in the W70 treatment in both years and in the W50 treatment in 2015. Although biomass was maintained in the W30 and FC treatments in 2015 and in the W50 and FC treatments in 2016, all other treatments experienced an overall reduction in biomass (Figure 1 and Figure S4). Maximum losses occurred in the extreme drought (Ext) and waterlogged treatments (WL) (Figure 1 and Figure S4). Biomass changes are not comparable across different plant tissues (Figure 1 and Figure S4). In the extreme drought treatments, the reduction in root biomass was proportionally less than the reduction in the aboveground tissues (leaves, twigs, stems), and in some cases (e.g., W30 in both years), decreased biomass in the aboveground tissues was accompanied by an increase in root biomass (Figure 1b and Table S1). Under waterlogged conditions (WL), the reduction of root biomass was proportionally less than the biomass reduction in aboveground tissues (Table S1). Over the course of the experiment, the variation pattern between the gradients is similar. The biomass increased at 20 days and 30 days, especially at 30 days under the treatment of FC and WL, and decreased at 20 days and 30 days under Ext and W10 (Figure 1a,b and Table S1).

Minimum values of TNSC occurred under W70. In general, TNSC values were higher with both increased and decreased soil water content (SWC) (Figure 2a and Figure S5a). However, there were significant differences in the response to drought between the 2015 and 2016 experiments in absolute terms ($p < 0.01$; Figure 2a): in 2015, maximum amounts of TNSC occurred in the W30 and W50 treatments and there was a decline with greater levels of water stress (W10 and Ext); in 2016, there was a progressive increase in the absolute amount of TNSC with maximum values in the Ext and W10 treatments.

The pattern of changes in SS (Figure 2b and Figure S6b and Table S2) basically mirrored that of TNSC, both with respect to the change relative to W70 and in terms of absolute differences between the two years. Maximum values of starch occurred in the W70 treatment and declined with both increased and decreased SWC (Figure 2c and Figure S6c and Table S2).

The variation of soluble sugar and starch concentrations in individual tissues was similar to that of the whole seedling. The minimum SS concentrations occurred in the W70 treatment, and concentrations increased with both increased and decreased SWC in all tissues (Figure 3a,c and Figure S6a,c). Similarly, maximum starch concentrations occurred in the W70 treatment and were reduced with both increased and decreased SWC in all tissues (Figure 3b,d and Figure S6b,d). The decreases in starch concentrations were, in general, progressive with increasing levels of drought or waterlogging. However, there was generally a progressive increase in SS concentrations in leaves, stems and roots with increasing levels of water stress, and the SS concentrations were significantly lower in both W70 treatments but similar in all other treatments (Figure 3a,c). There was a significant difference in the response to the Ext treatment between the 2015 and 2016 experiments with respect to leaf and stem SS concentrations (Figure S6a,c), but this difference was not manifested in the other tissues.

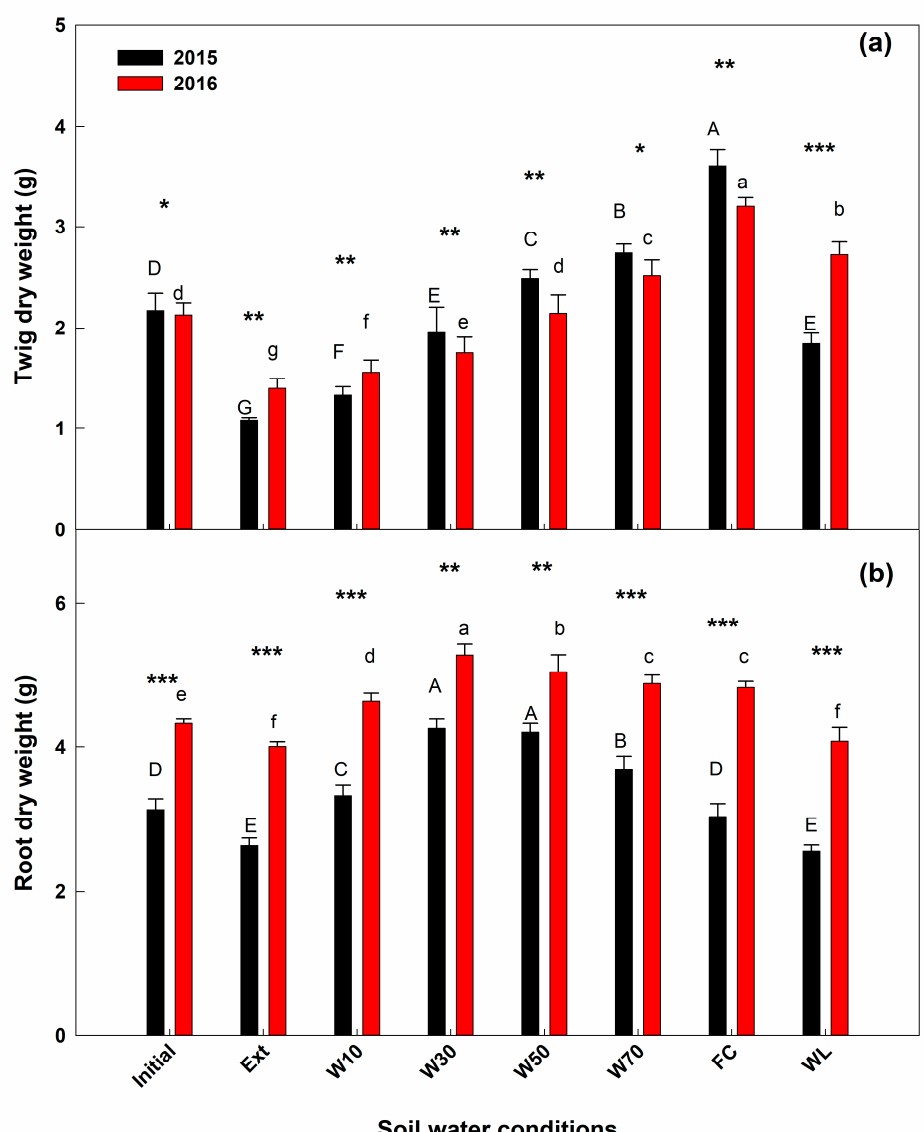

**Figure 1.** Changes in biomass in response to different water treatments. The plot shows the biomass at the beginning of the experiments (initial) and at the end of the 10-day treatments in 2015 (black bars) and 2016 (red bars) for (**a**) twigs and (**b**) roots. The treatments are Ext: extreme drought, W10: 10% of field capacity (FC), W30: 30% of FC, W50: 50% of FC, W70: 70% of FC, and WL: permanently waterlogged. Asterisks indicate a significantly different values (*, $p < 0.05$; **, $p < 0.01$; ***, $p < 0.001$) between 2015 and 2016. Different letters indicate significant differences between treatments in each year. The upper-case letter refers to 2015 and the lower-case letter refers to 2016.

The highest SS and starch concentrations were found in roots and leaves across all treatments (Figure 3c,d and Figure S6a,b). The increases in root SS compared to leaf SS were proportionally greater in seedlings under greater drought conditions than W70 (Tables S2 and S3). However, the increase in SS was proportionally larger in leaves than in roots in the WL treatment and the FC treatment in 2016. The reduction in starch concentrations in leaves was proportionally greater than the reduction in starch in roots under mild drought stress (W30 and W50), but it was of a comparable magnitude in the two components under extreme drought stress (W10 and Ext). The reduction in starch concentrations in leaves was proportionally greater than that of the reduction in roots under the FC and WL treatments (Figure 3d and Figure S6d).

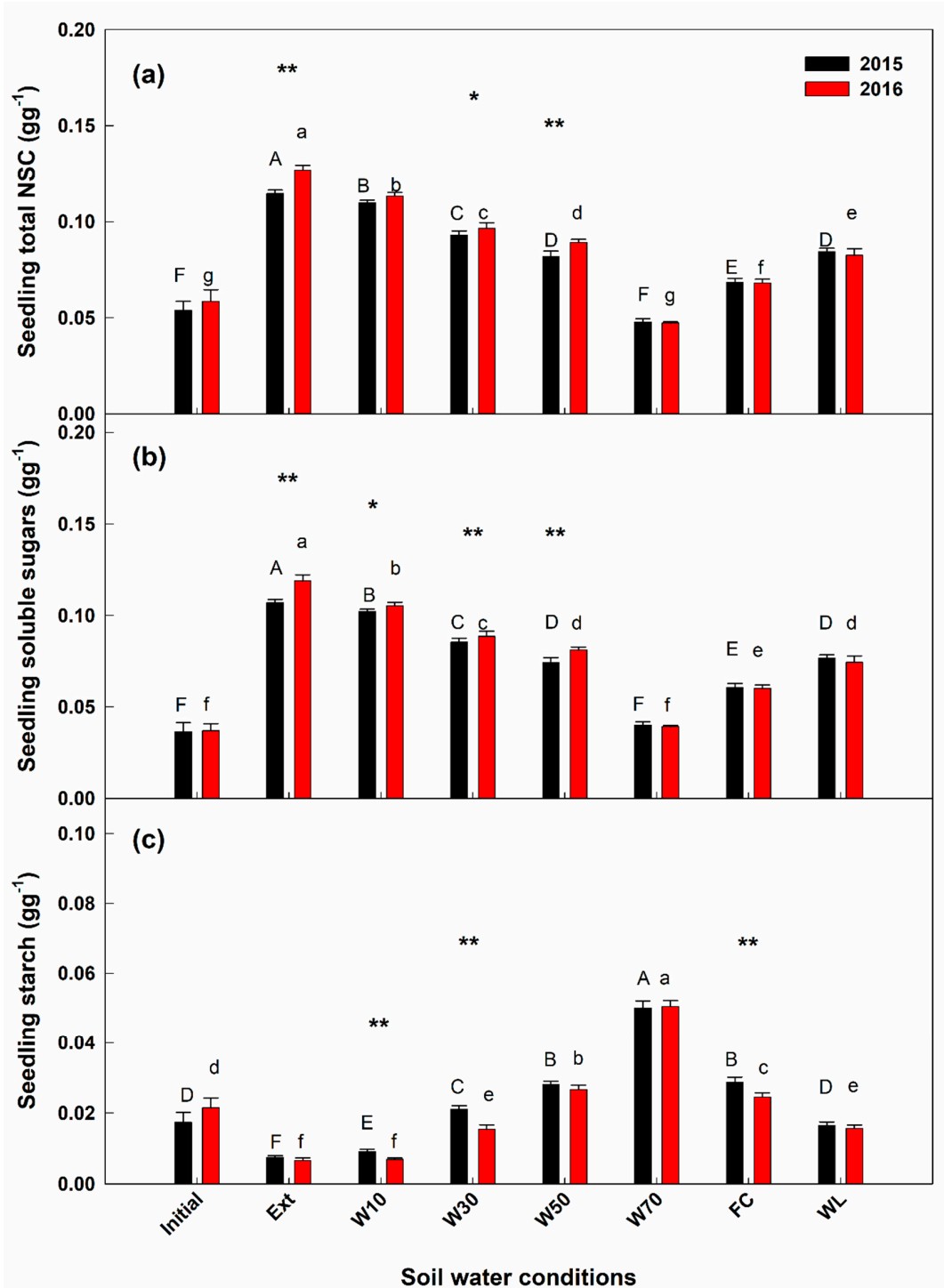

**Figure 2.** Changes in total non-structural carbohydrates (TNSC), soluble sugars (SS) and starch at the end of the 10-day water status treatments in 2015 (black bars) and 2016 (red bars), expressed as concentrations (g g$^{-1}$) and (**a**) stands for seedling TNSC, (**b**) for seedling SS and (**c**) for seedling starch. The treatments are Ext: extreme drought, W10: 10% of field capacity (FC), W30: 30% of FC, W50: 50% of FC, W70: 70% of FC, and WL: permanently waterlogged. Asterisks indicate a significantly different values (*, $p < 0.05$; **, $p < 0.01$; ***, $p < 0.001$) between 2015 and 2016. Different letters indicate significant differences between treatments in each year. The upper-case letter refers to 2015 and the lower-case letter refers to 2016.

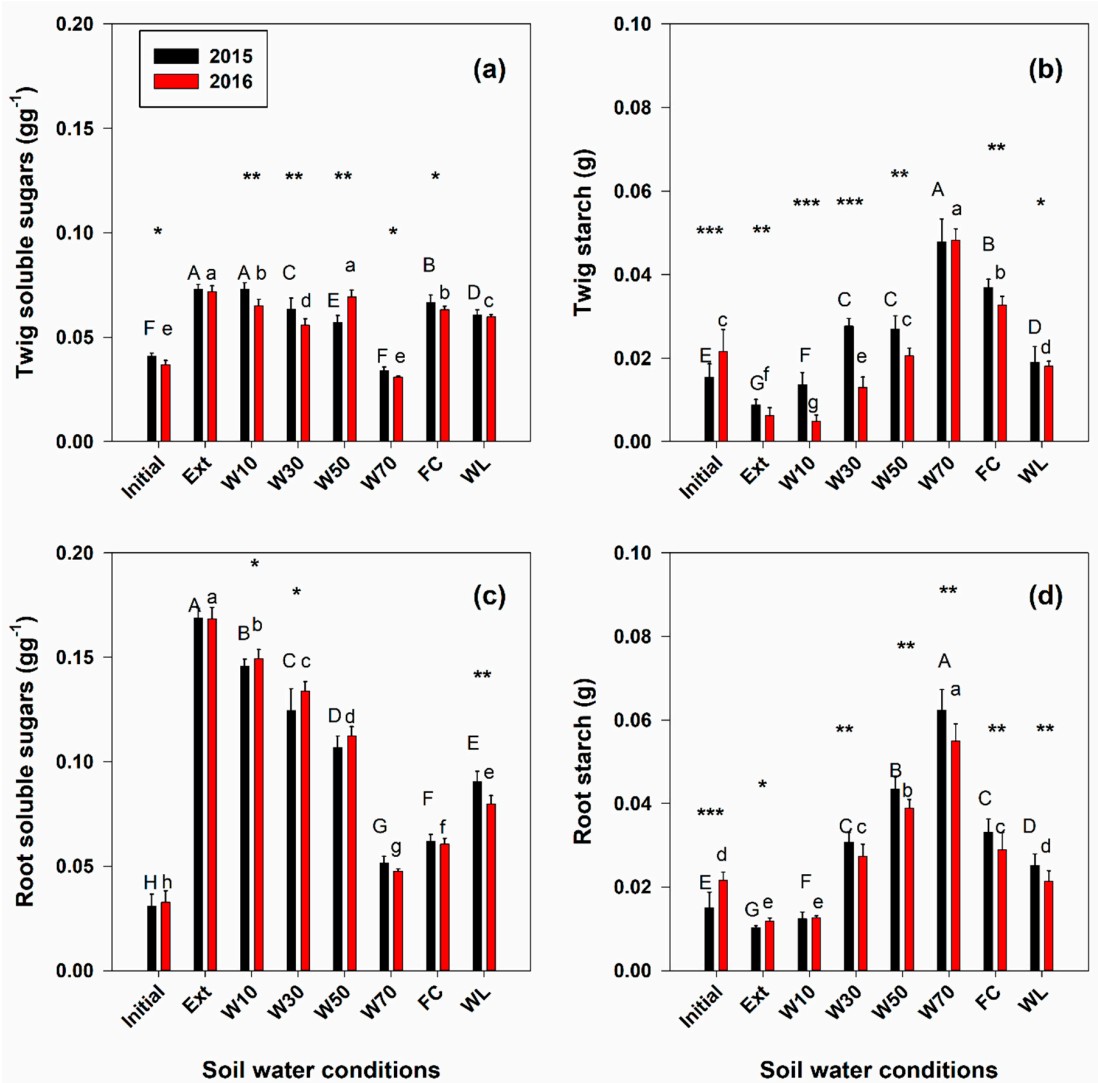

**Figure 3.** Soluble sugar (SS) and starch concentrations in twigs (**a**,**b**) and roots (**c**,**d**) for each of the water status treatments in 2015 (black bars) and 2016 (red bars). The treatments are Ext: extreme drought, W10: 10% of field capacity (FC), W30: 30% of FC, W50: 50% of FC, W70: 70% of FC, and WL: permanently waterlogged. Asterisks indicate a significantly different values (*, $p < 0.05$; **, $p < 0.01$; ***, $p < 0.001$) between 2015 and 2016. Different letters indicate significant differences between treatments in each year. The upper-case letter refers to 2015 and the lower-case letter refers to 2016.

The opposing trends of the changes in SS and starch are manifested in the ratio of SS to starch (Figure 4). This ratio is ca. 0.5 at the whole plant level when soil moisture was at 70% of field capacity (W70) but increases dramatically with both increased and decreased SWC (Figure 4a). The highest values occurred in the Ext and W10 treatments, with ratios of ca. 7 at the whole plant level. The value in the WL treatment was ca. 4.5. At the level of individual tissues, the most substantial changes in this ratio were observed in leaves under extreme drought: a ratio of >40 is found under extreme drought (Ext) and >20 is found in the W10 treatment (Figure 4b). In comparison, the SS/starch ratio in leaves under WL conditions was ca. 5. The changes in other tissues (roots, stems, twigs, Figure 4c–e) under extreme drought (Ext) were comparable (between ca. 13 and 18), but they were always larger than the increase in the SS/starch ratio under FC and WL treatments.

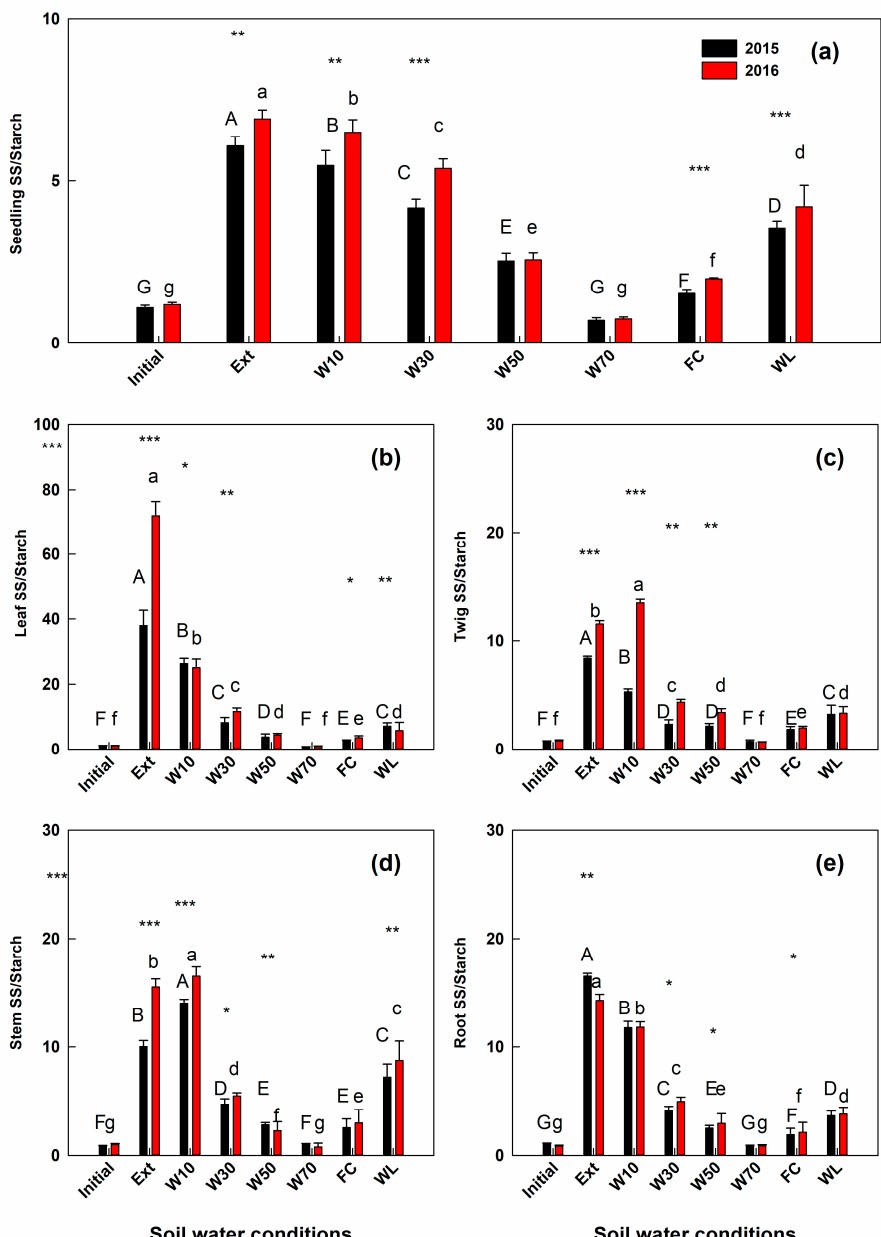

**Figure 4.** The ratio of soluble sugars (SS) to starch at the end of the 10-day water status treatments in 2015 (black bars) and 2016 (red bars) for (**a**) the whole plant (seedling SS/Starch); (**b**) leaves (leave SS/Starch); (**c**) twigs (twig SS/Starch); (**d**) stem (stems SS/Starch) and; (**e**) roots (root SS/Starch). Values were back-transformed averages, and error bars represent standard errors for the SS/starch of seedlings and tissues. The treatments are Ext: extreme drought, W10: 10% of field capacity (FC), W30: 30% of FC, W50: 50% of FC, W70: 70% of FC, and WL: permanently waterlogged. Asterisks indicate significantly different values (*, $p < 0.05$; **, $p < 0.01$; ***, $p < 0.001$) between 2015 and 2016. Different letters indicate significant differences between treatments in each year. The upper-case letter refers to 2015 and the lower-case letter refers to 2016. Note that the scale used for leaves is different from the scale used for twigs, stems and roots.

The concentrations of individual soluble sugars (fructose, glucose and sucrose) varied according to water treatments (Figure 5). The behaviour of fructose and sucrose was similar to that of SS overall: both showed a progressive increase with increasing drought and a less marked increase with increases in SWC compared to W70. This was apparent in all tissues except for twigs (Figure 5a,c), where the increase was modest across treatments, similar to the response of SS overall. Glucose concentrations

are much lower than both fructose and sucrose (Figure 5b). Glucose concentrations showed a marked increase under drought conditions (W30, W10 and Ext), particularly in leaves and roots (Figure 5b).

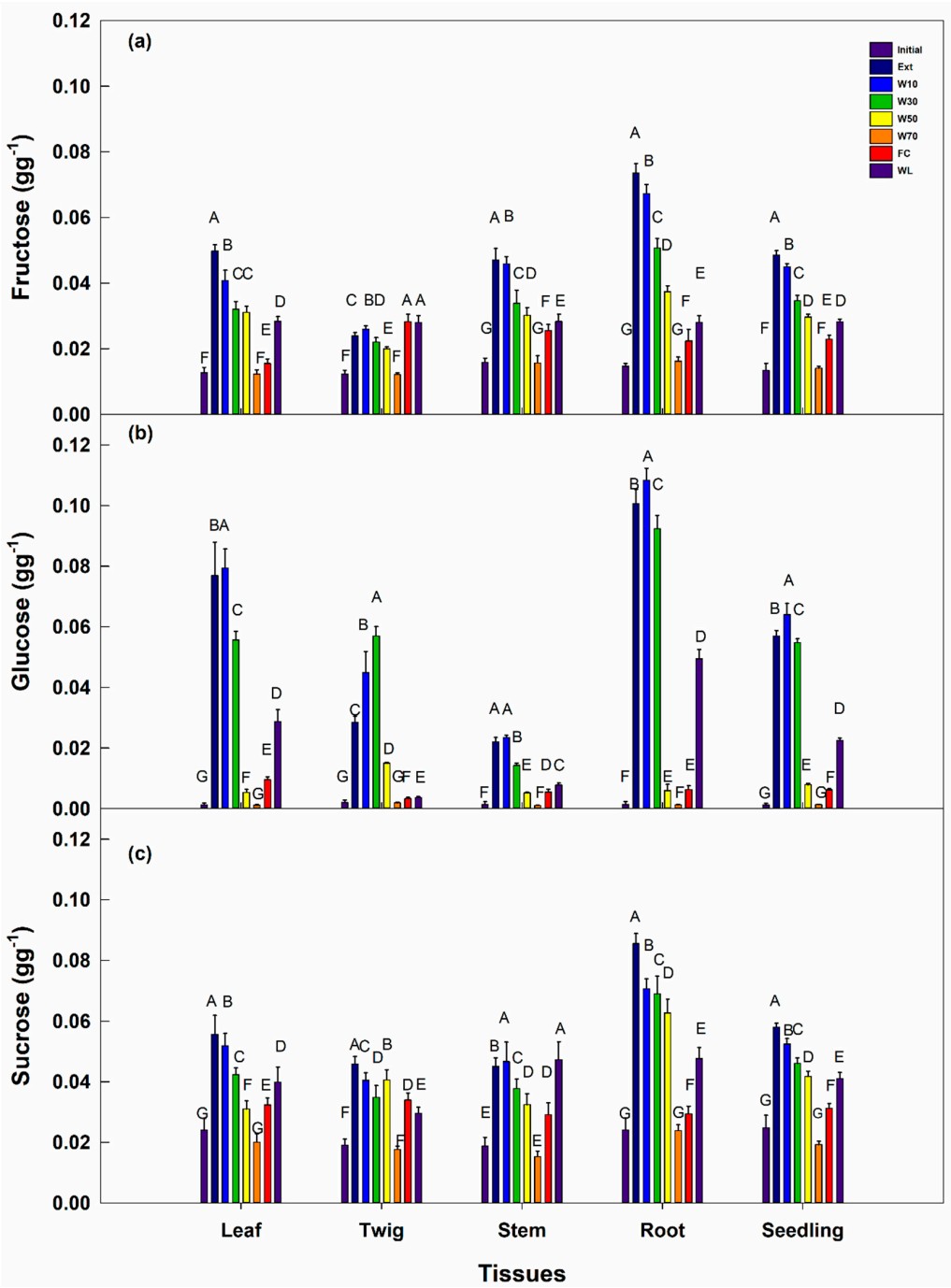

**Figure 5.** The mean concentrations of fructose (**a**), glucose (**b**) and sucrose (**c**) at the end of the 10-day water status treatments in 2015 and 2016. Values were back-transformed averages, and error bars represent standard errors for fructose, glucose and sucrose. The treatments are Ext: extreme drought, W10: 10% of field capacity (FC), W30: 30% of FC, W50: 50% of FC, W70: 70% of FC, and WL: permanently waterlogged. Asterisks indicate a significantly different value (*, $p < 0.05$; **, $p < 0.01$; ***, $p < 0.001$) between 2015 and 2016. Different letters indicate significant differences between treatments in each year. Note that the scale used for glucose is different from the scale used for fructose and sucrose.

In summary, over the course of the experiment, the variation of TNSC, SS and starch concentrations and pool sizes were similar in the two-year experiment, except that they were significantly reduced at 20 and 30 days under Ext and W10 treatments, especially at 30 days. The increase was intense at 10 days and then gradually decreased at 20 and 30 days under FC and WL treatments. Among them, the maximum value of starch appeared at 30 days under the W70 treatment, whereas the change in TNSC and SS was opposite that in starch, except that its minimum value appeared under the treatment of Ext and W10 at 30 days in 2015 and 2016.

## 4. Discussion

Our results indicated that the optimal growth conditions for *R. pseudoacacia* seedlings occurred when SWC was 70% (W70), resulting in increased biomass over the 30-day experiment and maximum concentrations of starch (and minimum concentrations of SS) in all plant tissues. The maximum concentration of starch (and minimal SS) is consistent with the idea that structural construction takes precedence over non-structural storage demands under optimal conditions [46]. Both reduced and increased SWC led to reduced biomass compared to the W70 treatment, accompanied by increased SS and decreased starch concentrations. Maximum biomass losses occurred under extreme drought and waterlogged conditions. However, the changes in biomass were proportionally less in roots than in any of the aboveground tissues under extreme drought stress, and the biomass of roots increased in cases of moderate drought stress (e.g., W30), even though total biomass decreased. This suggests that the seedlings were adapting to drought first by investing more carbon in belowground tissues to maximize water capture and that investment in roots is maintained preferentially even under severe drought stress (Ext). Investment in roots was also proportionally greater than investment in leaves under waterlogged conditions, although in this case, the reason is presumably to maximize oxygen uptake [47,48]. The overall reduction in biomass under drought and waterlogging conditions is consistent with previous studies [49–51] that showed similar changes in roots compared to aboveground biomass.

The TNSC changes under drought and waterlogging mirror the changes in SS; starch concentrations were generally very small, and thus changes in concentration have relatively little impact on TNSC concentrations. SS concentrations showed a progressive increase under both drought and increased SWC conditions. The increase in SS under stress is consistent with the idea of the maintenance of vital metabolic functions, including the maintenance of osmotic potential under stress conditions. Previous studies [7,31,52,53] have also shown that TNSC and SS increase under drought conditions. In our results, the concentrations of TNSC and SS were also increased under FC and WL treatments. There are some different underlying mechanisms between studies showing an increase under drought and under waterlogging. In addition, excessive increases in SS concentrations are probably not beneficial for plants in the long term [7]. Thus, the higher energy and no circulation in the plant-soil-atmosphere system can bring some risks for the plant itself; for example, some insects and microbes attack over time to gain access to the high SS concentrations [54]. However, it can clearly be beneficial in the short term as a way to maintain metabolic functions under stress [30,55,56].

Nevertheless, there is a disparity in the response to drought and waterlogging. The increase in SS in roots was proportionally greater than the increase in leaves under drought conditions, but under high SWC (e.g., FC, WL), the proportional increase was greater in leaves than in roots. This difference in response is also shown in the ratio of SS: starch, which increased more under drought stress than under waterlogging. The disparity is most obvious in the case of leaves under extreme drought stress (Ext). Although our experiments in this study were relatively short (e.g., 30 days), our results indicated significant changes in the ratio of SS and starch concentrations across all tissues; this contrasts with some previous studies on *R. pseudoacacia* [57] that argued that significant changes in stems and twigs only occurred after prolonged drought stress.

Despite the fact that the different SS are thought to have different metabolic roles [58], the changes in sucrose and fructose show similar patterns across the treatments and across tissues. Glucose

is present in comparatively small amounts under well-watered conditions but shows a marked increase under moderate (W10 and W30) and extreme drought (Ext) conditions in leaves and roots while twigs surprisingly show little change in TNSC or SS. Some previous studies merely showed twig traits in mature trees [59–61]. However, our results are basically consistent with the results of Zhang et al. [57] in the preliminary stage of drought; however, they did not conduct research under high SWC conditions [57]. Scientists previously investigated two conifer species under drought and found that glucose concentrations showed no significant differences in trend between control and drought treatments [62]. Rosas et al. [63] also investigated three woody species, and their results indicated a small change in TNSC [63], which is consistent with our results in this study.

The link between NSCs and osmotic adjustment or signal substances is at the core of our understanding of NSCs allocation in plants [11,50]. Although there are many theories about osmotic regulation and signal, its essential laws are still unclear. Soluble sugars are not only involved in the feedback regulation of photosynthetic rate, but also can cause changes in gene expression and enzyme activity [64]. According to our results, soluble sugar concentration was significantly increased under severe drought (Ext and W10), moderate drought (W30 and W50), field capacity (FC) and waterlogging conditions, especially during the initial stage (10-day). The main reason is that when plants encounter water stress, they can maintain the original cell expansion pressure to survive as quickly as possible, and then grow and develop. Under drought or waterlogged conditions, the imbalance between production and use leads to SS accumulation, which may mainly use SS distribution between different organs, thus increasing the transmission of important resistance signals to the roots under drought and stems under high soil water content. In response to stress, SS seems to be more important. Actually, starch also plays an important role. For example, Sulpice et al. (2009) showed that starch is a connecting point between stress conditions and photosynthesis, abscisic acid and reactive oxygen species, and starch is one of the main long-term stored carbohydrates in plants [64–66]. NSCs are important substances involved in plant life processes. In our study, the three components of soluble sugars present some similar response rules, but also have their own unique characteristics. For example, the trends of fructose are similar to sucrose, while glucose is the lowest. The concentrations of soluble sugars decreased in the order of sucrose > fructose > glucose. The main reason is that sucrose is not only the main form of carbohydrate transported in plants, but also can regulate cell metabolism at the level of gene expression [67,68]. Fructose and fructan are the main temporary storage forms of carbohydrate in plant nutrition tissue [69]. Glucose is the main connection in metabolisms. NSC concentration, pool, and distribution in plants reflect the overall carbon supply of plants and are key factors that determine the growth and survival of trees [70]. In our study, there are the physiological connotations of the asymmetric variations of soluble sugars and starch. Because plants are a complex system in face of drought and waterlogging, they can work together to resist water stress through other strategies, such as shifting in morphology, without using their own starch stock as much as possible.

*R. pseudoacacia* is a pioneer species in semi-arid environments. Its ability to survive moderate (and even relatively extreme) drought and waterlogging in the short term would be beneficial in such an environment, where both droughts and extreme rainfall events are frequent. Our experiments in this study may be too short to confirm that the utilization of NSCs would provide a buffer against prolonged extreme drought, although previous work indicates that NSC levels will increase over the growing season under drought conditions and that even larger changes can occur in stems and twigs on this 30-day short time scale [57]. While increased levels of NSCs under stress clearly enhance plant resilience, the application of our findings at an ecosystem level should be done with caution as our study has some limitations due to the relatively short periods of measurements (only 30 days) [70].

Process-based models play an important role in investigating and quantifying the impacts of climate and climate variability on tree growth and carbon allocations at different time scales [11,32,58,71–76]. Unfortunately, there are very few process-based carbon models that are able to simulate NSC allocations for different components of trees under both drought and waterlogging conditions. Our study suggests an urgent need to explicitly measure NSC pools in different plant

tissues and incorporate them into process-based tree growth and carbon cycle models to capture the complex response of these NSC pools to different types of hydraulic stress.

## 5. Conclusions

Drought and waterlogging stress had significant impacts on NSC concentrations: starch decreased and SS increased in response to increased stress. Our results reveal that individual tree components of SS showed different responses to drought and waterlogging across tissues. Under drought conditions, the increase in the SS concentrations in roots was larger than the increase in SS concentrations in leaves, consistent with buffering to maximize water uptake. Root biomass increased under moderate drought stress, despite decreases in aboveground biomass, consistent with the idea that changes in NSC concentration help maximize water uptake. Under waterlogged conditions, however, the increase in the SS concentrations in leaves was larger than the increase in SS concentrations in roots. The strength of the response is asymmetric, with the largest increase occurring under extreme drought. These allocation changes are broadly compatible with the roles of individual SS under different conditions. The results showed that plants adopt different acclimatization strategies under different types of water stress. However, changes in the NSC pools were limited in twigs and stems, suggesting that there was only a limited need to preserve the functioning of transport tissues under short-term water stress.

This study provides unique observations and valuable information about the impacts of water stress on the total amount of NSCs, NSC allocation between soluble and insoluble components and NSC distribution across different plant tissues. To address questions about the allocation mechanisms of non-structural carbohydrates of different tree species in response to drought and waterlogging, it will be important to consider how and how much of the NSC is used to alleviate stress and sustain plant function. It will also be important to examine the response to different frequencies and different lengths of stress periods in future studies.

**Supplementary Materials:** The following are available online at http://www.mdpi.com/1999-4907/9/12/754/s1, Figure S1: Air temperature in both years, Figure S2: Air humidity in both years, Figure S3: The greenhouse experimental designs, Figure S4: Changes in biomass in response to different water treatments, Figure S5: Changes in NSC pools in response to different water treatments, Figure S6: Soluble sugars and starch concentrations of leaf and stem, Table S1: Biomass in different tissues at the 20-day and 30-day of the experiments and at the end of the soil moisture treatments, respectively, Table S2: Percentage change in soluble sugars (SS) and starch across treatments, Table S3: Effects of water status on gravimetric soil moisture across treatments, Table S4: Effects of water status on leaf relative water content across treatments.

**Author Contributions:** B.Y., C.P. and M.W. designed the experiment; B.Y. carried out the greenhouse experiments; B.Y. performed the experiments in the lab; B.Y. analysed data; B.Y. wrote the manuscript. B.Y., C.P., S.P.H., H.W., H.W., Q.Z. and M.W. revised and improved the manuscript.

**Funding:** This study was financially supported by the National Key R&D Program of China (2016YFC0500203), the Qian Ren program, the Natural Sciences and Engineering Research Council of Canada (NSERC) Discovery Grant, the China State Administration of Foreign Expert Affairs as a High End Foreign Expert at Northwest A&F University, Yangling, the ERC-funded project GC2.0 (Global Change 2.0: Unlocking the past for a clearer future, grant number 694481), the National Natural Science Foundation of China (41571081), the National Key R&D Program of China (2016YFC0501804), the National Natural Science Foundation of China (41601098) and the Young Excellence Program for the Teachers of College of Forestry, Northwest A&F University (Z111021603).

**Acknowledgments:** Many thanks to Yi Zhang for helping implement our experiments and carry out seedling culture and water control in the greenhouse and Weiming Yan for helping analyse NSCs and strengthen our discussion. We are also indebted to Mingxi Du and Lin Jiang for their guidance with statistical analysis.

**Conflicts of Interest:** The authors declare no conflict of interest.

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
