# Peer review of "Allocation Mechanisms of Non-Structural Carbohydrates of Robinia pseudoacacia L. Seedlings in Response to Drought and Waterlogging"

_forests, doi:10.3390/f9120754_

Round 1
Reviewer 1 Report
This paper is very thorough and appears to illustrate some differences between NSCs under drought and waterlogged situations.
There is an editing flaw which sees the results and discussion appear before the methods.
The results must be summarised and the extra material should go into the supplimentary material. The first paragraph of the results is excellent and so is the last paragraph of the results but the rest must be summarised. There are far too many graphs illustrated in the paper. Exemplar graphs should be used and the remaining graphs included in the supplimentary material. For example in Figure 1 should show twig and root biomass only, the others are simmilar and should not be used. 10 graphs should be the maximum used 2 per figure. In the figure 2 caption you don't say what each graph is - a, b, c, d, e, f they are used in the figure but not explained in the caption. What is the difference between 2015 and 2016 data? Did you just run the experiment again? Why?
The discussion is very short. What are the full implications of these results? Are they relevant for other species? Why did this species give this result?
The work is worthwhile but the results section must be reduced and the discussion expanded.
Thank you.
Author Response
Response to Reviewer 1 Comments
Point 1: There is an editing flaw which sees the results and discussion appear before the methods.
Response: We are sorry for this editing flaw and we corrected this and put the methods before the results and discussion in the revised MS.
The results must be summarized and the extra material should go into the supplementary material. The first paragraph of the results is excellent and so is the last paragraph of the results but the rest must be summarized. There are far too many graphs illustrated in the paper. Exemplar graphs should be used and the remaining graphs included in the supplementary material. For example in Figure 1 should show twig and root biomass only, the others are similar and should not be used. 10 graphs should be the maximum used 2 per figure. In the figure 2 caption you don't say what each graph is - a, b, c, d, e, f they are used in the figure but not explained in the caption.
Response: Thanks for your good suggestions and have revised graphs and put the extra material into the supplementary material as suggested. Yes, we show twig and root biomass only for the Fig. 1. We have revised the figure captions, including figure 2 caption. Asterisks indicate a significantly different values (*, P < 0.05; **, P < 0.01; ***, P < 0.001) between 2015 and 2016. Different letters indicate significant differences between treatments in each year. The upper-case letter refers to 2015 and the lower-case letter refers to 2016.
Point 2: What is the difference between 2015 and 2016 data? Did you just run the experiment again? Why?
Response: Good questions! Yes, 2015 and 2016 are repeated experiments, which was expected to increase the reliability and repeatability of our study.
Point 3: The discussion is very short. What are the full implications of these results? Are they relevant for other species? Why did this species give this result? The work is worthwhile but the results section must be reduced and the discussion expanded.
Response: Good questions! The full implications of our results are that NSCs can be used as indicators to evaluate the response mechanisms of plants under stress conditions. Our main goal is to explore NSCs allocation and their potential biological significance of Robinia pseudoacacia L. (black locust) seedlings in short-term but explicitly considering the potential functions of different NSC components.
Our main research object is the black locust seedlings, and the black locust belongs to a pioneer species of drought resistance in arid and semi-arid areas of China. Our results may be relevant for broadleaf species, but the R pseudoacacia L. has the characteristics of leguminosae. For example, rhizobium is involved in resisting drought stress which is our ongoing work in the next step.
Thanks for your good suggestion! Yes, we have reduced the results section by deleting some information in the results and put extra material into the supplementary material as Appendix and added more information in the discussions section (Please see lines 340-368).

Reviewer 2 Report
The work described in the present manuscript has reasonable value. The experimental design is adequate and the coverage over different drought stress levels extensive. However, the manuscript offers little more outside the non-structural carbohydrate measurements, and the discussion is weak. Before the manuscript can be accepted for publication, there are some points that, in my opinion, authors must address.
1. The information covered in Figure 1 and Table S1 seem to be the same, and the data regarding biomass production at 20 and 30 days seems to be missing. In addition, even though it is explain in the materials and methods, authors should specify in the figures that the biomass is expressed as Dry weight, as it makes it easier for the reader to understand the data (the same is applicable to Figures 2 and 3).
2. The meaning of the symbols showing the statistical significance in the figures is not clear. For example, in Figure 1 asterisks are explained to show differences with respect to W70, but then there are asterisks above the bars of the W70 treatment. Authors should try to clarify their use of symbols.
3. In my opinion, biomass production on its own does not provide enough information to understand the physiological status of the plants. Considering that the work is focused on drought stress, authors should provide information about the water status of the plants: preferably their water potential, but at least they should include the plant relative water content.
4. The values for the non-structural carbohydrate content of the plants at the beginning of the experiment ("Initial" treatment) should also be included in Figures 2, 3 and 4.
5. The discussion must be improved, since the paper is fairly descriptive as it is. For example, authors could expand on the role of sugars as osmoprotectants, the roles of different sugars, or the physiological connotations of the asymmetric variations of soluble sugars and starch.
These are some minor points that also require the authors' attention:
L61. This sentence is not clear. Consider rephrasing.
L88. Authors shouldn´t use the term "obviously", since in heavy draught conditions is not rare that the biomass is maintained or even decreased.
L150. What does "ca." stand for?
L219. I don´t think that it´s correct to claim that "SS goes down to roots under drought stress and up to the aboveground material under excess water". Since SS are synthetized in the aerial part, it´s most likely a consequence of a decrease in transport out of the aboveground tissues rather than an transport up from the roots. Consider rephrasing.
L280. I guess that authors mean "buckets" instead of "bucks".
L282 to 287. This sentence is confusing. Consider rephrasing.
Author Response
Response to Reviewer 2 Comments
Dear Editor,
We like to submit our revised MS entitled “Allocation Mechanisms of Non-structural Carbohydrates of Robinia pseudoacacia Seedlings in Response to Drought and Waterlogging” (Manuscript ID: forests-386212) to forests.
First of all, we would like to express our great appreciation to you and two reviewers for your valuable suggestions and comments on the previous version of the manuscript. The revised manuscript has been improved as a result of your constructive advices.
Our responses to the reviewer’s comments and modifications are detailed in following pages. We hope that the revised manuscript is satisfactory to your journal. Please feel free to contact me if further information is required.
Thank you very much for your consideration. I am looking forward to hearing from you soon.
Sincerely yours,
Bin Yang
Center for Ecological Forecasting and Global Change,
College of Forestry, Northwest A&F University,
Yangling, Shaanxi 712100, China
Reviewer #2:
The work described in the present manuscript has reasonable value. The experimental design is adequate and the coverage over different drought stress levels extensive. However, the manuscript offers little more outside the non-structural carbohydrate measurements, and the discussion is weak. Before the manuscript can be accepted for publication, there are some points that, in my opinion, authors must address.Comments and Suggestions for Authors
RE: Thank you for your positive feedback! We tried our best to address your questions and concerns in the followings:
Point 1: The information covered in Figure 1 and Table S1 seem to be the same, and the data regarding biomass production at 20 and 30 days seems to be missing. In addition, even though it is explain in the materials and methods, authors should specify in the figures that the biomass is expressed as Dry weight, as it makes it easier for the reader to understand the data (the same is applicable to Figures 2 and 3).
Response: Good suggestions. We agree with you! We have added "20-day" and “30-day” data in Table S1. And the same is applicable to Figures 2 and 3, this study mainly shows the NSC allocation of 10-day, which does not involve time dynamics.
Point 2: The meaning of the symbols showing the statistical significance in the figures is not clear. For example, in Figure 1 asterisks are explained to show differences with respect to W70, but then there are asterisks above the bars of the W70 treatment. Authors should try to clarify their use of symbols.
Response: Good point! We have revised the figure captions and clearly explained the meaning of the symbols showing the statistical significance. The asterisks above the bars are explained to show differences between 2015 and 2016.
Point 3: In my opinion, biomass production on its own does not provide enough information to understand the physiological status of the plants. Considering that the work is focused on drought stress, authors should provide information about the water status of the plants: preferably their water potential, but at least they should include the plant relative water content.
Response: Thanks for the good suggestions. We agree with you! Unfortunately, we don’t measure water potential. As suggested, we have added the leaf relative water content in the Table S4 as:
Table S4. Effects of water status on leaf relative water content across treatments, where Ext is extreme drought (no watering), W10, W30, W50 and W70 are at 10%, 30%, 50% and 70% of field capacity (FC), respectively and WL is waterlogged. The values are means ± SE.
Point 4: The values for the non-structural carbohydrate content of the plants at the beginning of the experiment ("Initial" treatment) should also be included in Figures 2, 3 and 4.
Response: Good suggestions and agree. We have added the "Initial" treatment in Figures 2, 3, 4 and 5, as suggested.
Point 5: The discussion must be improved, since the paper is fairly descriptive as it is. For example, authors could expand on the role of sugars as osmoprotectants, the roles of different sugars, or the physiological connotations of the asymmetric variations of soluble sugars and starch.
Response: Thank you for your suggestions. We have expanded our discussion on lines 340-368 as followings:
“The link between NSCs and osmotic adjustment or signal substances is at the core of our understanding of NSCs allocation in plants (Kozlowski 2002, Dietze 2014). Although there are many theories about osmotic regulation and signal, its essential laws are still unclear. Soluble sugars are not only involved in the feedback regulation of photosynthetic rate, but also can cause changes in gene expression and enzyme activity (Ramel 2009). According to our results, soluble sugar concentration was significantly increased under severe drought (Ext and W10), moderate drought (W30 and W50), field capacity (FC) and waterlogging conditions, especially during the initial stage (10-day). The main reason is that when plants encounter water stress, they can maintain the original cell expansion pressure to survive as quickly as possible, and then grow and develop. Under drought or waterlogged conditions, the imbalance between production and use leads to SS accumulation, which may mainly use SS distribution between different organs, thus increasing the transmission of important resistance signals to the roots under drought and stems under high soil water content. In response to stress, SS seems to be more important. Actually, starch also plays an important role. For example, Sulpice et al (2009) showed that starch is a connecting point between stress conditions and photosynthesis, abscisic acid and reactive oxygen species, and starch is one of the main long-term stored carbohydrates in plants (Ramel 2009, Sulpice 2009, Rudack 2017). NSCs are important substances involved in plant life processes. In our study, the three components of soluble sugars present some similar response rules, but also have their own unique characteristics. For example, the trends of fructose are similar to sucrose, while glucose is the lowest. The concentrations of soluble sugars decreased in the order of sucrose> fructose > glucose. The main reason is that sucrose is not only the main form of carbohydrate transported in plants, but also can regulate cell metabolism at the level of gene expression (Kühn 2011, Watanabe 2016). Fructose and fructan are the main temporary storage forms of carbohydrate in plant nutrition tissue (Chardon 2013). Glucose is the main connection in metabolisms. NSC concentration, pool, and distribution in plants reflect the overall carbon supply of plants and are key factors that determine the growth and survival of trees (Sheen 2014). In our study, there are the physiological connotations of the asymmetric variations of soluble sugars and starch. Because plants are a complex system in face of drought and waterlogging, they can work together to resist water stress through other strategies, such as shifting in morphology, without using their own starch stock as much as possible.”
These are some minor points that also require the authors' attention:
L61. This sentence is not clear. Consider rephrasing.
Response: Good suggestions. Yes, we have rewrote this sentence.
L88. Authors shouldn´t use the term "obviously", since in heavy draught conditions is not rare that the biomass is maintained or even decreased.
Response: Thank you for your suggestions. We have deleted the term "obviously" in lines 166.
L150. What does "ca." stand for?
Response: Good questions! Yes, "ca." stands for circa, means approximately.
L219. I don´t think that it´s correct to claim that "SS goes down to roots under drought stress and up to the aboveground material under excess water". Since SS are synthetized in the aerial part, it´s most likely a consequence of a decrease in transport out of the aboveground tissues rather than a transport up from the roots. Consider rephrasing.
Response: We agree with you! We have deleted the sentence "SS goes down to roots under drought stress and up to the aboveground material under excess water" in lines 312-313.
L280. I guess that authors mean "buckets" instead of "bucks".
Response: Yes, revised as suggested.
L282 to 287. This sentence is confusing. Consider rephrasing.
Response: We have rephrased this sentence.

Round 2
Reviewer 2 Report
The authors should be congratulated for their hard work addressing all my comments. However, it looks like the uploaded version of the file containing the supplementary material is the same as in the original submission (the file is dated 24-10-2018), which doesn't contain any of the new figures they added (Figures S4, S5 and S6; and Table S4). Please provide the updated Supplementary material. Once the new file is available, I consider that the manuscript is suitable for publication.
Author Response
Dear Editor,
We like to submit our revised MS entitled “Allocation Mechanisms of Non-structural Carbohydrates of Robinia pseudoacacia Seedlings in Response to Drought and Waterlogging” (Manuscript ID: forests-386212) to forests.
First of all, we would like to express our great appreciation to you and two reviewers for your valuable suggestions and comments on the previous version of the manuscript and supplementary material. The revised supplementary material has been improved as a result of your constructive advices.
Our responses to the reviewer’s comments and modifications are detailed in following pages. We hope that the revised supplementary material is satisfactory to your constructive advices. Please feel free to contact me if further information is required.
Thank you very much for your consideration. I am looking forward to hearing from you soon.
Sincerely yours,
Bin Yang
Center for Ecological Forecasting and Global Change,
College of Forestry, Northwest A&F University,
Yangling, Shaanxi 712100, China
Response to Reviewer 2 Comments-Round 02
(x) I would not like to sign my review report
( ) I would like to sign my review report
English language and style
( ) Extensive editing of English language and style required
( ) Moderate English changes required
(x) English language and style are fine/minor spell check required
( ) I don't feel qualified to judge about the English language and style
Yes | Can be improved | Must be improved | Not applicable | |
Does the introduction provide sufficient background and include all relevant references? | (x) | ( ) | ( ) | ( ) |
Is the research design appropriate? | (x) | ( ) | ( ) | ( ) |
Are the methods adequately described? | (x) | ( ) | ( ) | ( ) |
Are the results clearly presented? | (x) | ( ) | ( ) | ( ) |
Are the conclusions supported by the results? | (x) | ( ) | ( ) | ( ) |
Comments and Suggestions for Authors
RE: We appreciate your positive feedbacks on our MS, and will try our best effort to the questions in the followings:
The authors should be congratulated for their hard work addressing all my comments. However, it looks like the uploaded version of the file containing the supplementary material is the same as in the original submission (the file is dated 24-10-2018), which doesn't contain any of the new figures they added (Figures S4, S5 and S6; and Table S4). Please provide the updated Supplementary material. Once the new file is available, I consider that the manuscript is suitable for publication.
Response: We are sorry for this submitting flaw and we corrected this and resubmitted the updated supplementary material, including traced and cleared versions. Please see the new uploaded files.
Submission Date
24 October 2018
Date of this review
21 Nov 2018 12:47:02
